# The DNA Damage Response Is Differentially Involved in HPV-Positive and HPV-Negative Radioresistant Head and Neck Squamous Cell Carcinoma

**DOI:** 10.3390/cancers13153717

**Published:** 2021-07-23

**Authors:** Marieke Bamps, Rüveyda Dok, Sandra Nuyts

**Affiliations:** 1Laboratory of Experimental Radiotherapy, Department of Oncology, KU Leuven, University of Leuven, 3000 Leuven, Belgium; marieke.bamps@kuleuven.be (M.B.); ruveyda.dok@kuleuven.be (R.D.); 2Department of Radiation Oncology, Leuven Cancer Institute, UZ Leuven, 3000 Leuven, Belgium

**Keywords:** head and neck cancer, radioresistance, HPV

## Abstract

**Simple Summary:**

Head and neck cancers can be divided in two major groups according to their risk factors, being high-risk human papillomavirus related (HPV-positive) and alcohol and tobacco related (HPV-negative) head and neck cancers. The majority of the locally advanced patients are treated with radiotherapy. However, up to 50% of these patients show local recurrences. The majority of these recurrences are linked to resistance to radiotherapy treatment. It is known that the response to DNA damage, also a process called the DNA damage response, is an important factor that determines the effectivity of radiotherapy. Here, we assessed the role of the DNA damage response in the resistance process to radiotherapy of head and neck cancers, by generating head and neck cancer cells resistant to radiotherapy. We show that the DNA damage response is differentially involved in the resistance process of HPV-positive and HPV-negative head and neck cancer cells. More specifically, HPV-positive radiotherapy-resistant cells showed increased ability to repair the DNA damage induced by radiotherapy. HPV-negative radiotherapy-resistant cells showed increased capacity to replicate after radiotherapy treatment. Despite this difference, inhibition of the DNA damage response enhanced the effect of radiotherapy in both groups.

**Abstract:**

Radioresistance is a major cause of recurrences and radiotherapy (RT) failure in head and neck squamous cell carcinoma (HNSCC). DNA damage response (DDR) is known to be important for RT response, but its role in radioresistance is not fully understood. Here, we assessed the role of DDR in the radioresistance process of HNSCC by generating radioresistant clones from both HPV-positive SCC154 and HPV-negative SCC61 cells. We show that fractionated RT decreased RT response of HPV-positive and HPV-negative radioresistant clones in vitro and in vivo. Moreover, HPV-positive and HPV-negative radioresistant clones were characterized by differential DDR response. HPV-positive radioresistant clones showed less residual double-strand break damage and increased G2/M arrest recovery after RT, indicating an acquisition of increased DDR kinetics. In contrast, HPV-negative radioresistant clones showed less micronucleated cells after RT and increased survival upon checkpoint inhibition, indicating an increased replicative capacity. Inhibiting key factors of DDR in combination with RT rescued the radioresistant phenotype of both HPV-positive and HPV-negative radioresistant clones. Altogether, our results not only highlight the importance of DDR response in the radioresistance process of HPV-positive and HPV-negative HNSCC, but also provide possibilities for new therapies for HNSCC patients in recurrent settings.

## 1. Introduction

With approximately 600,000 new cases yearly, head and neck squamous cell carcinoma (HNSCC) is the sixth most common cancer worldwide [1]. The two main carcinogenic routes causing HNSCC are infection with the human papillomavirus (HPV) and the abuse of alcohol and tobacco consumption, defined as HPV-positive and HPV-negative cancers respectively [1]. Although HPV-positive HNSCC patients have better prognosis after treatment, the majority of all locally advanced unresectable HNSCC patients are treated with chemo-radiotherapy (RT) [2,3]. 

Up to 26% and 10% of patients with locally advanced HPV-negative and HPV-positive HNSCC, respectively, are affected with locoregional recurrences after treatment [4,5]. Radioresistance is one of the main reasons behind tumor relapse seen in these patients [6,7]. Although radiobiology has been extensively studied, the underlying mechanisms of radioresistance after fractionated RT are not fully understood [8]. Activation of the DNA damage response (DDR), one of the cornerstones of radiobiology, is detrimental for survival after RT [9,10]. In line with this, we and others have shown the importance of DDR in the RT response of both HPV-positive and HPV-negative HNSCC [11,12,13,14,15]. In addition, the radiosensitizing potential of DDR inhibition has been demonstrated in several preclinical models of HNSCC, suggesting that DDR could be an important factor for radioresistance [11,12,15,16,17]. 

Here, we investigate the importance of DDR in a radioresistant setting for both HPV-positive and HPV-negative HNSCC. Radioresistant cells were generated through fractionated irradiation to mimic patient relapse in vitro, since matched primary and recurrent tumor samples are unavailable to study the mechanisms underlying radioresistance. Next, we elucidated the role of DDR by targeted inhibition in both HPV-positive and HPV-negative radioresistant cells. 

## 2. Materials and Methods

### 2.1. Cell Lines and Reagents

All used cell lines were authenticated and cultured as previously described in [11]. HPV-positive HNSCC cell line SCC154 was purchased from the German collection of micro-organisms and cell cultures (DSMZ, Braunschweig, Germany) and cultured in Minimal Essential Medium (MEM, Thermo Fisher Scientific, Waltham, MA, USA) supplemented with 10% fetal bovine serum (FBS, Thermo Fisher Scientific), 1% L-glutamine (Thermo Fisher Scientific) and 1% non-essential amino acids (Thermo Fisher Scientific). HPV-negative HNSCC cell line SCC61 was gifted by Dr. A. Begg from the Netherlands Cancer Institute and cultured in Dulbecco’s Modified Eagle Medium (DMEM, Thermo Fisher Scientific) supplemented with 10% FBS and 1% sodium pyruvate (Thermo Fisher Scientific). Cell lines were incubated at 37 °C and passaged via trypsinization. Cell lines were authenticated with short-tandem repeat profiling by ATCC. All experiments were performed with mycoplasma-free cells.

Inhibitors ABT-888, AZD6738, AZD7762, LY2603618 and NU7441 were purchased from Selleck Chemicals (Houston, TX, USA). Inhibitors were dissolved in dimethylsulfoxide (DMSO, Sigma-Aldrich, Saint-Louis, MO, USA).

The in vitro irradiation experiments were performed with Baltograph (199 kV photons, 15 mA, Balteau NDT, Oupeye, Belgium) and Small Animal Radiation Research Platform (220 kV photons, 13 mA, X-strahl, Camberley, UK). The in vivo irradiation experiments were performed with Small Animal Radiation Research Platform (220 kV photons, 13 mA, X-strahl).

### 2.2. Establishment of Isogenic Radioresistant HNSCC Cells

Established HNSCC cell lines were treated with a fractionated RT treatment schedule of 2 Gy per fraction given two times a week, up to a total dose of 60 Gy. Between fractions and after completion of the treatment schedule, recovery periods of 2–3 days and 10–12 days were implemented, respectively. In addition, cells were subcultured independent of fractionated irradiation, preventing the risk of radiosensitive cells overgrowing radioresistant cells as described in [18]. Clonal selection was performed to isolate radioresistant clones by limited dilution assays, as previously described in [19]. Afterwards, the RT response of the generated clones was assessed by a short-term survival assay, the sulforhodamine B (SRB) assay described previously in [13]. More specifically, clones were seeded together with their parental cells in 96-well plates and after an attachment period of 24 h, irradiated with doses of 0, 3 and 6 Gy. One week after irradiation, all cells were fixed with trichloroacetic acid (100810, Millipore, Burlington, MA, USA) and stained with sulforhodamine B (Sigma-Aldrich, Saint-Louis, MO, USA). Thereafter, 10 mM Tris-base solution (Trizma, T1503, Sigma-Aldrich, Saint-Louis, MO, USA) was added to release the SRB and absorbance at 570 nm was measured. Survival fractions (SF) were determined by comparing the optical values as a measure of survival of irradiated cells compared to the survival of non-irradiated controls. To define the increase in radioresistance, relative survival of the clones at 3 and 6 Gy was compared to the relative survival of parental cells at 3 and 6 Gy. A clone was defined as radioresistant when the relative RT response, SF, compared to the parental cells increased at least 1.2-fold and 1.8-fold at 3 and 6 Gy, respectively. For HPV-negative cell line SCC61, radioresistant clones R61-1 and R61-4 were selected for further experiments. For HPV-positive cell line SCC154, radioresistant clones R154-1 and R154-4 were selected for further experiments. Clonogenic assays were performed to validate radioresistance. Survival curves were fitted to a linear-quadratic (LQ) model (Sf = e^(−αD−βD²)^).

### 2.3. Clonogenic Assay

Cells were seeded and after attachment overnight, treated with DMSO (0 µM) or indicated concentration of drug 2 h before exposure to increasing RT doses of 0, 2, 4 and 6 Gy. Drug concentrations were chosen based on previous studies and with no/limited baseline toxicities [11]. Medium was refreshed 24 h after drug exposure. Cells were fixed after 2–3 weeks with 2.5% glutaraldehyde in PBS and stained with 0.4% crystal violet (Sigma-Aldrich, Saint-Louis, MO, USA). Colonies containing 50 cells or more were counted with Gelcount (Oxford Optronix, Abington, UK). Dose enhancement factors (DEF) at RT dose of 2 Gy were calculated for each inhibitor by dividing the SF of RT alone (0 µM) by the SF of RT with inhibitor. 

### 2.4. Crispr-Cas9

The CRISPR-Cas9 screen was performed as previously reported in [11]. More specifically, two crRNAs were designed with the CRISPR tool from Zhang lab (MIT, Cambridge, MA, USA) for each gene to exclude off-target effects. As negative control, a scrambled crRNA was used. Cells were transfected with Lipofectamine 2000 (Invitrogen, Carlsbad, CA, USA) according to the IDT Alt-RT CRISPR-Cas9 protocol. Transfection efficiency was determined with Block-it and was above 70% for all cell lines. Forty-eight hours after transfection, cells were treated with a RT dose of 3 and 6 Gy or untreated. Survival of cells was assessed 1 week after RT treatment with SRB assay. The survival rate of irradiated cells with crRNA-Cas9 complexes was normalized to the survival rate of non-irradiated control.

### 2.5. Doubling Time Assay

Cells were seeded in 96-well plates and fixed from day 1 until day 7 via SRB assay. The population doubling time was determined in the exponential growth phase with the following formula (with *DT* = doubling time, *T* = the incubation time in any unit, *Xe* = the cell number at the end of the incubation time, *Xb* = the cell number at the beginning of the incubation time). 

$$DT=T\frac{ln2}{l_{n}\frac{Xe}{Xb}}$$

### 2.6. Cell Cycle Analysis

Cells were fixed with 70% ethanol and stained with 10 µg/ml propidium iodide (Sigma-Aldrich, Saint-Louis, MO, USA) containing 100 µg/mL RNase A (Invitrogen, Carlsbad, CA, USA). For Bromodeoxyuridine (BrdU) analysis, cells were incubated with 75 µM 5’-Bromo-2’-Deoxyuridine (Sigma-Aldrich, Saint-Louis, MO, USA) for 2 h and further processed via BD Biosciences protocol (Anti-BrdU B44). Cell cycle distribution was assessed with BD FACSVerse (Piscataway, NJ, USA).

### 2.7. Immunoblotting

Proteins were extracted with RIPA buffer (150 mM sodium chloride, 1% Triton X-100, 0.50 sodium deoxycholate, 0.10% SDS and 50 mM tris pH 8.0) containing protease and phosphatase inhibitors (Roche, Basel, Switzerland). Protein concentrations were determined using Bradford assay (Bio-rad, Hercules, CA, USA) with albumin bovine serum (Sigma-Aldrich, Saint-Louis, MO, USA). Ten-15 µg of protein was loaded onto SDS-PAGE (Invitrogen, Carlsbad, CA, USA). The following antibodies were used: pATM S1981 (13050S CST, Danvers, MA, USA), ATM (2873S CST, Danvers, MA, USA), pATR S428 (28538 CST, Danvers, MA, USA), ATR (27903 CST, Danvers, MA, USA), pCHK2 T68 (2197 CST, Danvers, MA, USA), CHK2 (3440 CST, Danvers, MA, USA), pCHK1 S345 (2348 CST, Danvers, MA, USA), pCHK1 S296 (2349 CST, Danvers, MA, USA), CHK1 (2360 CST, Danvers, MA, USA), pH3 S10 (53348 CST, Danvers, MA, USA), pDNApk S2056 (Abcam, Cambridge, MA, USA), DNApk total (12311S CST, Danvers, MA, USA), PAR (Enzo Life Sciences, Farmingdale, NY, USA), Vinculin (Sigma-Aldrich, Saint-Louis, MO, USA) and beta-actin (CST, Danvers, MA, USA). Protein bands were detected using enhanced chemiluminescence (ECL, PerkinElmer, Waltham, MA, USA) and visualized with AI680 Western blot imager (GE Healthcare, Chicago, IL, USA). Densitometry was performed via Image J. All protein values were corrected to their loading control and phosphorylated protein values were corrected to their total protein values. Relative values were calculated by dividing each value by the control (untreated) value.

### 2.8. Immunofluorescence

Cells were seeded in µClear 96-well plates (Greiner Bio-one, Kremsmünster, Austria) and fixed with 4% paraformaldehyde. After permeabilization with 0.3% Triton X-100 (Sigma-Aldrich, Saint-Louis, MO, USA) in 0.5% BSA-PBS, cells were stained with primary antibody against pH2AX Ser139 (JBW301, Millipore, Burlington, MA, USA) followed by staining with 488 Alexa fluor secondary antibody (4408 CST, Danvers, MA, USA). Nuclei were counterstained with DAPI (D9542, Sigma-Aldrich, Saint-Louis, MO, USA). For micronuclei analysis, cells were fixed as described above and stained with DAPI (D9542, Sigma-Aldrich, Saint-Louis, MO, USA). Immunofluorescence images were acquired with In Cell analyzer 2000 (GE Healthcare, Chicago, IL, USA). 

### 2.9. In Vivo Xenograft Models

HPV-positive and HPV-negative parental and radioresistant cells were subcutaneously injected in both flanks of female nu/nu NMRI (Janvier Labs, Le Genest-Saint-Isle, France) mice. Experiments were performed according to the ethical committee of KU Leuven (P163/2017). Mice were divided in a control group and RT group. Mice from the latter group were treated with fractionated RT with 2 Gy per fraction for 3 or 5 consecutive days when tumors reached a volume of approximately 100 mm^3^. Tumor volumes were determined with caliper measurements. Body weight and health status of the mice were monitored daily.

### 2.10. Statistical Analysis

Statistical significance was tested using one-way ANOVA, two-way ANOVA and extra-sum of squares F-test using Graphpad prism 6.01 software (GraphPad Prism Software Inc., San Diego, CA, USA). *p*-values < 0.05 were considered statistically significant.

## 3. Results

### 3.1. Fractionated RT Results in Radioresistance in Both HPV-Positive and HPV-Negative HNSCC

To assess the underlying mechanisms of radioresistance, we generated isogenic radioresistant clones R61-1 and R61-4 from HPV-negative cell line SCC61, and R154-1 and R154-4 from HPV-positive cell line SCC154, by fractionated RT with 2 Gy per fraction up to total dose of 60 Gy. Compared to parental cells SCC154 and SCC61, fractionated RT significantly increased the relative clonogenic survival of R154-1, R154-4, R61-1 and R61-4 clones, with 1.6-, 1.2-, 1.6- and 1.4-fold increase at 2 Gy, respectively (Figure 1A). Although we could detect differences in the clonogenic growth of non-irradiated conditions between SCC154 and SCC61 and their derived radioresistant R154-1 and R61-4 clones, this difference in clonogenic growth was not detected in R154-4 and R61-1 clones (Appendix AA). 

Next, we assessed whether we could detect the decrease in radiosensitivity also in vivo conditions by investigating the delay in tumor regrowth after RT treatment in HPV-positive and HPV-negative radioresistant xenografts compared to their parental xenografts (Figure 1B,C and Appendix AB,C). Tumor volume doubling (TVD) times were assessed, since these reflect growth delay after RT in vivo, and the probabilities by Kaplan–Meier statistics were determined. TVD was defined as the time interval between the tumor volume at the start of treatment and the time when tumors reached twice their initial tumor volume. Both HPV-positive and HPV-negative radioresistant models showed a decreased tumor growth delay after RT compared to the respective parental models (Figure 1B,C and Appendix AB,C). 

In line with the in vitro data, HPV-positive radioresistant models showed higher variation in RT response compared to HPV-negative radioresistant models. Fifty percent of the mice reached a TVD at day 5 and day 15 for R154-1 and R154-4 respectively, while 50% of SCC154 in vivo xenografts reached TVD at day 19, indicating a tumor growth delay of at least 4 days compared to the radioresistant clones. In addition, HPV-negative radioresistant clones showed an increase in radioresistance in vivo, with 50% of the R61-1 and R61-4 xenografts reaching TVD at day 21 and day 22 respectively. In contrast, 50% of SCC61 xenografts reached TVD at day 45, indicating a tumor growth delay of 23 days. 

We also determined the TVD in non-irradiated conditions for all the in vivo models (Figure 1B,C and Appendix AB,C). Although radioresistant tumors showed a faster tumor growth compared to parental tumors, this was only significant for HPV-positive models. To see whether we could also detect the differences in vitro, we determined the doubling times of parental and radioresistant clones (Appendix AD). HPV-negative radioresistant cells showed a significant difference, with an average increase of 4 h in the proliferation rate compared to HPV-negative SCC61 parental cells. Although HPV-positive radioresistant cells showed also an average increase of 4 h in the proliferation rate, this did not reach significance. 

### 3.2. HPV-Positive and HPV-Negative Radioresistant Cells Show Differential Involvement of the DNA Damage Response 

Several studies demonstrated that fractionated RT can result in alterations in DDR [20,21,22]. To elucidate the role of DDR in the radioresistance process of the generated radioresistant cells, we first assessed the DDR kinetics by determining γH2AX foci as double-strand break (DSB) damage marker and cell cycle distribution. Four hours after RT, around 2-fold induction of H2AX phosphorylation was detected in both HPV-positive and HPV-negative parental cells and all derived radioresistant clones (Figure 2A and Appendix A and Appendix A). 

In line with previous studies [13,14,23,24,25,26], HPV-positive SCC154 cells were characterized with residual DNA damage, as shown by significant increased levels of γH2AX, even at 28 h after RT. In contrast, γH2AX levels at 28 h in the HPV-positive radioresistant R154-1 and R154-4 clones were 1.3-fold lower in comparison to parental cells (Figure 2A and Appendix A). To investigate whether the faster γH2AX kinetics seen in HPV-positive radioresistant cells was due to differences in baseline expression levels of DDR sensors, protein activation status via phosphorylation was determined. The absence of differences in baseline protein expression levels of DNA damage sensors pATM and pATR and effectors pCHK2 and pCHK1 in HPV-positive SCC154 and the derived radioresistant clones (Figure 2B) confirmed faster DNA repair kinetics after RT in HPV-positive radioresistant clones. In concordance with this, a significantly faster G2/M arrest recovery at 32 h until 72 h after RT was seen in HPV-positive radioresistant clones compared to parental cells (Figure 2C and Appendix A). This further confirms faster cell cycle recovery in HPV-positive radioresistant clones compared to parental cells.

HPV-negative SCC61 cells and radioresistant clones showed a similar decrease in γH2AX foci levels at 16 h and 24 h after RT, indicating the absence of residual DSB damage and suggesting the absence of differential DDR after RT (Figure 2A and Appendix A). No differences in baseline activation of DDR sensors pATM and pATR and their effector proteins pCHK2 and pCHK1 were detected between HPV-negative radioresistant clones and parental cells (Figure 2B), further confirming the absence of DDR effects.

This was verified by similar cell cycle distribution after RT, with similar G2/M arrest induction at 12 h and similar recovery of G2/M arrest after RT in both HPV-negative radioresistant clones and parental cells (Figure 2C and Appendix A).

Since increased ability of cells to divide and survive with RT-induced DNA damage is a characteristic of radioresistant cells [9], we assessed the percentage of micronucleated cells (Figure 2D and Appendix A) and phosphorylated H3 (pH3) expression levels (Appendix AA) after RT in HPV-positive and HPV-negative parental cells and their derived radioresistant clones. RT induced 33% and 26% micronucleated cells in both HPV-positive SCC154 and HPV-negative SCC61 parental cells after 72 h, respectively. No difference in the percentage of micronucleated cells was detected between HPV-positive radioresistant clones and parental cells. In addition, HPV-positive radioresistant R154-1 and R154-4 clones showed a relative increase of 1.7 and 3.1-fold in pH3 expression levels compared to the parental SCC154 cells (Appendix AA), indicating an increase in mitotic cells. These results are in concordance with cell cycle distribution, with HPV-positive radioresistant clones showing faster cell cycle recovery. 

In contrast, HPV-negative radioresistant clones showed significantly less micronucleated cells compared to SCC61 parental cells at 48 h and 72 h after RT (Figure 2D and Appendix A). Hence, this suggests fewer mitotic defects and lower levels of genomic instability in the HPV-negative radioresistant clones. In addition, HPV-negative radioresistant clones R61-1 and R61-4 showed a relative increase of 22.0- and 30.0-fold pH3 expression levels compared to SCC61 cells 48 h after RT (Appendix AA). These results verify that radioresistant clones show an increased replicative ability after RT. 

ATR is an important regulator of S-phase progression and G2/M cell cycle checkpoint activation after replication-based DNA damage and genomic instability [27]. Moreover, ATR inhibition is known to induce micronuclei and mitotic cell death, especially in cells with increased genomic instability [27]. We hypothesized that alterations in ATR cell checkpoint regulations could explain the lower levels of micronucleated cells detected in HPV-negative radioresistant clones after RT. Therefore, we first assessed whether we could determine the sensitivity to ATR inhibition in both the HPV-positive and HPV-negative parental cells and their radioresistant clones by using ATR inhibitor AZD6738. ATR inhibition showed a limited effect on survival in HPV-positive SCC154 cells and derived radioresistant clones (Figure 2E). In addition, inhibition of downstream ATR targets CHK1 and CHK2 with CHK1 and CHK1/2 inhibitors showed significant differences in survival fractions between HPV-positive parental cells and HPV-positive radioresistant clones at certain concentrations (Appendix AB). However, these differences in survival fractions between parental cells and radioresistant clones were not consistent. In contrast, ATR inhibition and even inhibition of downstream ATR targets CHK1 and CHK2 with CHK1 and CHK1/2 inhibitors resulted in significantly lower survival fractions of HPV-negative SCC61 parental cells compared to radioresistant clones, confirming that HPV-negative radioresistant cells are characterized by lower levels of mitotic defects (Figure 2E and Appendix AC) [28]. To verify the altered cell cycle regulation seen in HPV-negative radioresistant clones, we investigated the differences in S-phase progression by assessing the BRDU-incorporation after RT recovery. In line with our previous results, HPV-negative radioresistant clones showed a significantly lower BRDU-incorporation 24 h after RT compared to parental cells, confirming that HPV-negative radioresistant clones spent less time in the S-phase after RT recovery (Appendix AD), indicating replication problems after RT in parental cells. These results, together with the absence of a difference in DSB repair kinetics and decreased number of micronuclei, indicate that fractionated RT altered the intrinsic capacity of R61-1 and R61-1 cells to cope and survive with RT-induced DNA damage.

### 3.3. Targeting DDR Results in Increased Radiosensitization of HPV-Positive and HPV-Negative Radioresistant Cells

Our previous study showed that inhibition of DDR resulted in radiosensitization of both HPV-positive and HPV-negative HNSCC models [11]. Therefore, we assessed whether inhibition of key players in DDR could enhance RT response in radioresistant cells. For this we compared the clonogenic survival of the radioresistant clones upon targeted inhibition in combination with RT to the survival fraction of their parental cells. The selection of targets was based on our previous study, in which we showed that DSB break repair pathway non-homologous end joining (NHEJ), single-strand break repair pathway base excision repair (BER) and cell cycle were important targets for radiosensitization of both HPV-positive and negative HNSCC. Inhibition of CHK1/2 with AZD7762, inhibition of DNA-PKcs with NU7441 and inhibition of BER with ABT-888 in combination with RT resulted in radiosensitization of radioresistant HNSCC cells, to the extent that they rescued the radioresistant phenotype (Figure 3A). 

In line with our previous study, inhibition of NHEJ resulted in the highest level of radiosensitization in radioresistant cells with an average DEF_2Gy_ of 3.0 for both HPV-positive and HPV-negative radioresistant cells. Inhibition of cell cycle checkpoints CHK1/2 resulted in average DEF_2Gy_ of 1.5 for all radioresistant cells, whereas inhibition of BER showed the least radiosensitizing potential with average DEF_2Gy_ of 1.3. In addition, Western blot analysis showed targeted inhibition of all drugs (Figure 3B). Next, we investigated the radiosensitizing potential of targeted DDR inhibition with CRISPR-Cas9 mediated knockdown of the selected genes by short-term survival assay (Appendix A). Radioresistant clones showed higher relative survival compared to parental cells in all the conditions. The RT response upon knockdown was heterogeneous among the radioresistant clones; however, the overall results were in line with the drug inhibition data.

## 4. Discussion

The absence of matched samples from primary and recurrent tumors from patients treated with RT makes isogenic cell lines generated by fractionated RT an ideal model to study processes causing radioresistance [18,21,22,29,30,31,32,33]. Here, we developed radioresistant HNSCC cells from established HNSCC cell lines, originated from treatment-naive tumors [34,35]. 

The focus of current in vitro studies is on generating and understanding the biology of radioresistance in the HPV-negative subgroup, mainly due to the high locoregional relapse rates in HPV-negative HNSCC [20,21,22,31,33,36]. 

To have a broad understanding of radioresistance within different subgroups of HNSCC, we selected a representative HPV-negative HNSCC cell line, SCC61, and a representative HPV-positive HNSCC cell line, SCC154. Fractionated RT given in clinically relevant fractions of 2 Gy induced radioresistance in both selected cell lines. To the best of our knowledge, we are the first to develop HPV-positive radioresistant HNSCC cells.

Recently, it has been shown that fractionated RT results in a heterogeneous group of clonal populations with heterogeneous RT responses compared to their parental cell line [20,22]. To reduce the possible heterogeneity, we selected two radioresistant clones for our further experiments. Although all radioresistant clones showed a decreased RT response in both in vitro and in vivo conditions, the HPV-negative R61-1 and R61-4 clones showed a more radioresistant phenotype compared to their parental cells. Moreover, the HPV-positive clones were characterized by a more diverse response to RT in vitro and in vivo compared to HPV-negative radioresistant clones, indicating the possibility of a different mechanism behind radioresistance. 

Wu et al. showed that genes involved in DDR were upregulated in esophageal squamous cell carcinoma (ESCC) radioresistant cells and tumor biopsies of esophageal cancer patients, highlighting the importance of DDR in the radioresistance process. Moreover, the overlap of dysregulated pathways and genes between ESCC cells and tumor samples from recurrent ESCC patients show the potential of using in vitro radioresistant models for determining underlying processes of radioresistance [37]. In line with this, we detected an increased DDR capacity in HPV-positive radioresistant clones compared to parental cells, suggesting an increased ability to cope with RT-induced DNA damage. Less DNA damage induction and increased DSB resolving in radioresistant HNSCC cells is in concordance with previous studies [20,21,22,38,39]. In contrast to HPV-positive HNSCC, we could not detect a difference in DDR kinetics between HPV-negative radioresistant clones and the parental cells. 

The ATR pathway is known to prevent replication stress and mitotic catastrophe by functioning on downstream cell cycle checkpoints [27]. Differences in sensitivity to checkpoint inhibition are suggested to be dependent on intrinsic differences in DNA replication [40,41]. HPV-negative radioresistant clones showed lower levels of mitotic defects after RT and were less reliant on the ATR pathway for baseline survival compared to parental cells, suggesting decreased genomic instability and increased intrinsic replicative ability in HPV-negative radioresistant clones. 

Interestingly, radioresistant cells generated from the SCC61 cell line by Bansal et al. [21] showed differential DDR kinetics when compared to parental cells, with radioresistant cells showing less γH2AX foci and G2/M cell cycle arrest after RT. Although the different fractionated RT schedules and the absence of clonal selection may explain the difference compared to our results, it also highlights the presence of different radioresistance mechanisms even in the same cell lines. 

Several studies have shown that DDR inhibition resulted in radiosensitization of HPV-positive and HPV-negative HNSCC using established cell lines [11,12,15,16,17]. In addition to these studies, we have investigated the radiosensitization potential of DDR inhibitors in HPV-positive and HPV-negative isogenic radioresistant cells. DDR inhibition resulted in radiosensitization of radioresistant cells independent of HPV status and detected DDR differences. 

Although we were able to generate radioresistant cell lines from both HPV-positive and HPV-negative groups of HNSCC, we only used a single cell line for each group to investigate radioresistance. Expanding the number of isogenic radioresistant cells derived from additional HPV-positive and HPV-negative HNSCC cell lines and validating these models with patient-derived tumor xenografts will increase the clinical relevance. Another limitation of this study is that we only investigated the role of DDR in the radioresistance process. Although we have shown that DDR was differentially involved in HPV-positive and HPV-negative radioresistant clones, these differences were small and the driving forces of radioresistance were not unraveled. Further studies investigating the changes at the genomic, transcriptomic and epigenetic level could further broaden the understanding of the processes driving radioresistance and the involvement of DDR therein.

## 5. Conclusions

In conclusion, we have shown that DDR is differentially involved in the radioresistant process of HPV-positive and HPV-negative HNSCC cells. HPV-positive radioresistant cells showed an increased DNA repair capacity, whereas HPV-negative radioresistant cells showed an increased replicative capacity. Despite these differences, combining DDR inhibitors together with RT renders both HPV-positive and HPV-negative radioresistant cells more sensitive to RT, offering possibilities for new therapies for patients with recurrent HNSCC, a population with limited therapeutic options.

## Figures and Tables

**Figure 1 cancers-13-03717-f001:**
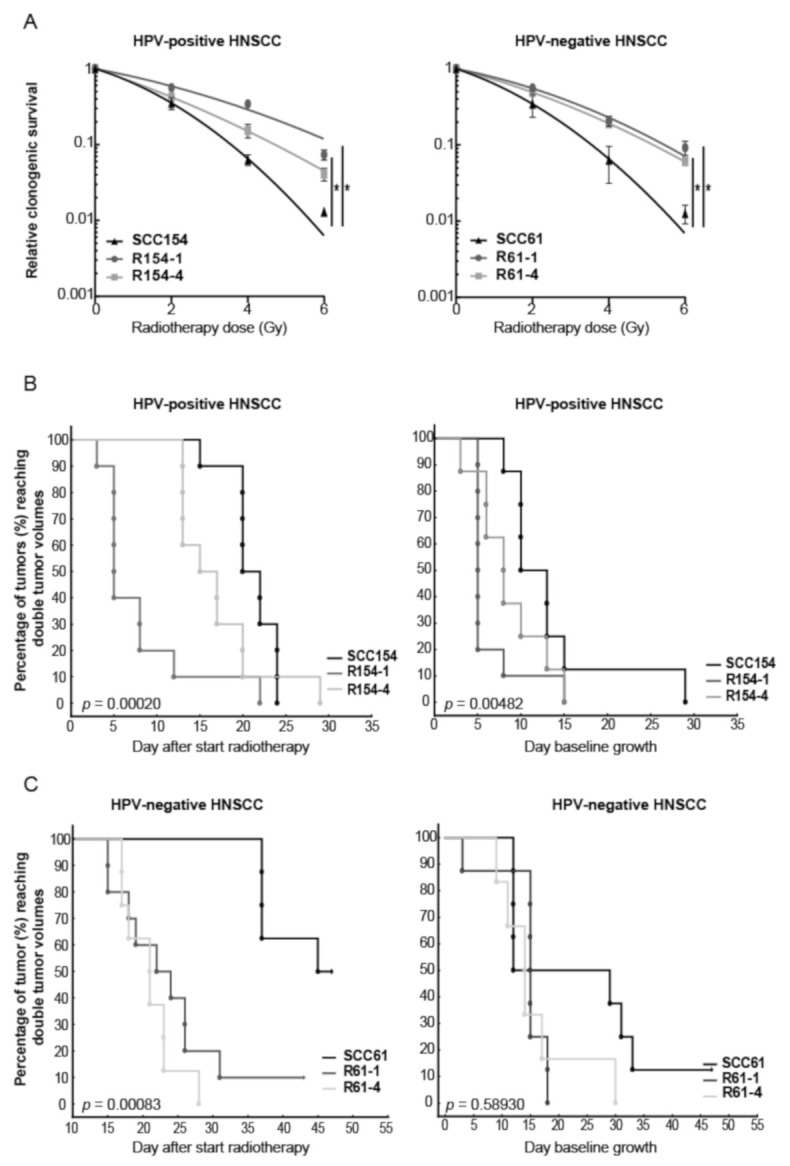
HPV-positive and HPV-negative HNSCC clones showed increased radioresistance in vitro and in vivo. (**A**) Clonogenic survival of HPV-positive and HPV-negative radioresistant clones was compared to their parental cell lines. Data are represented as the mean ± SEM for *n* = 3. Survival curves are fitted via the linear quadratic model. * *p*-values < 0.05 were calculated by extra sum-of-squares F-test. (**B**,**C**) Kaplan–Meier curves of HPV-positive and HPV-negative parental and radioresistant in vivo xenografts showed the percentage of tumors reaching double tumor volume. RT dose was given in a fractionated manner with 2 Gy fraction for 3 and 5 consecutive days for HPV-positive and HPV-negative models, respectively. All HPV-positive irradiated and unirradiated groups consisted of 10 tumors. HPV-negative SCC61, R61-1 and R61-4 irradiated groups contained 8, 10 and 8 tumors respectively. HPV-negative SCC61, R61-1 and R61-4 unirradiated groups contained 8, 8 and 6 tumors respectively. Statistics were performed by chi-square test.

**Figure 2 cancers-13-03717-f002:**
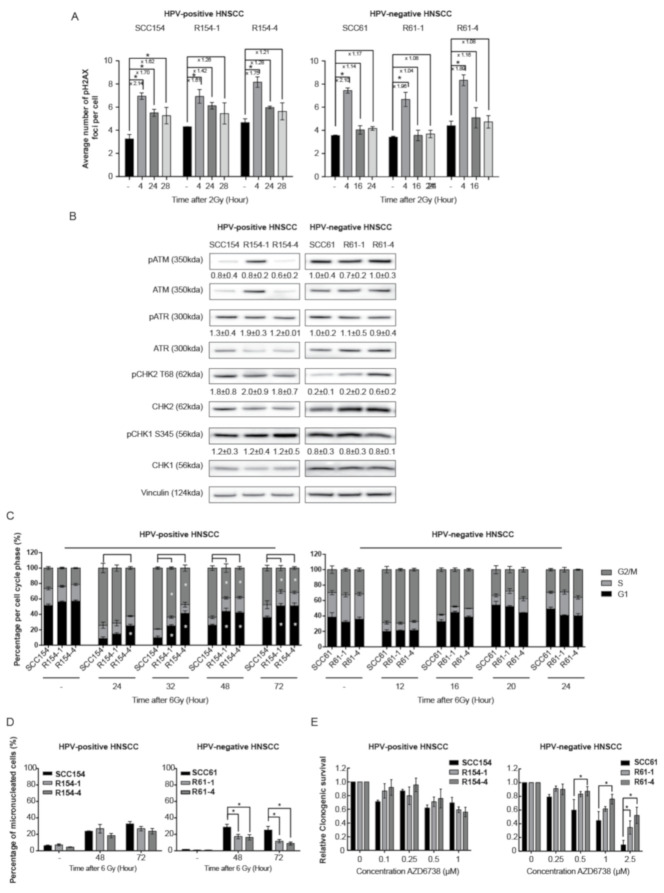
HPV-positive and HPV-negative radioresistant HNSCC clones show differences in the involvement of the DNA damage response. (**A**) Average number of γH2AX foci per cell was determined in untreated cells (−) and at indicated time points after a RT dose of 2 Gy. (**B**) Baseline protein levels of pATM (350kDa), ATM (350kDa), pATR (300kDa), ATR (300kDa), pCHK2 T68 (62kDa), CHK2 (62kDa), pCHK1 S345 (56kDa), CHK1 (56kDa) and Vinculin (124kDa) in HPV-positive and HPV-negative parental and radioresistant cells. KDa is the molecular weight as determined by the protein standard. (**C**) Cell cycle distribution (percentage %) in G1, S and G2/M phase was determined at different time points after a RT dose of 6 Gy. (**D**) The percentage of cells with micronuclei was determined at indicated time points after a RT dose of 6 Gy and in unirradiated cells (−). (**E**) Relative survival was determined after treatment with indicated concentrations (µM) of ATR inhibitor AZD6738. Relative survival was determined by means of clonogenic survival and is shown as the mean ± SEM clonogenic survival fraction of drug treated conditions relative to vehicle treated control cells, *n* = 3. (**A**–**E**) Data are presented as the mean ± SEM for *n* = 3. * *p*-values < 0.05 were calculated with ANOVA with multiple comparisons test.

**Figure 3 cancers-13-03717-f003:**
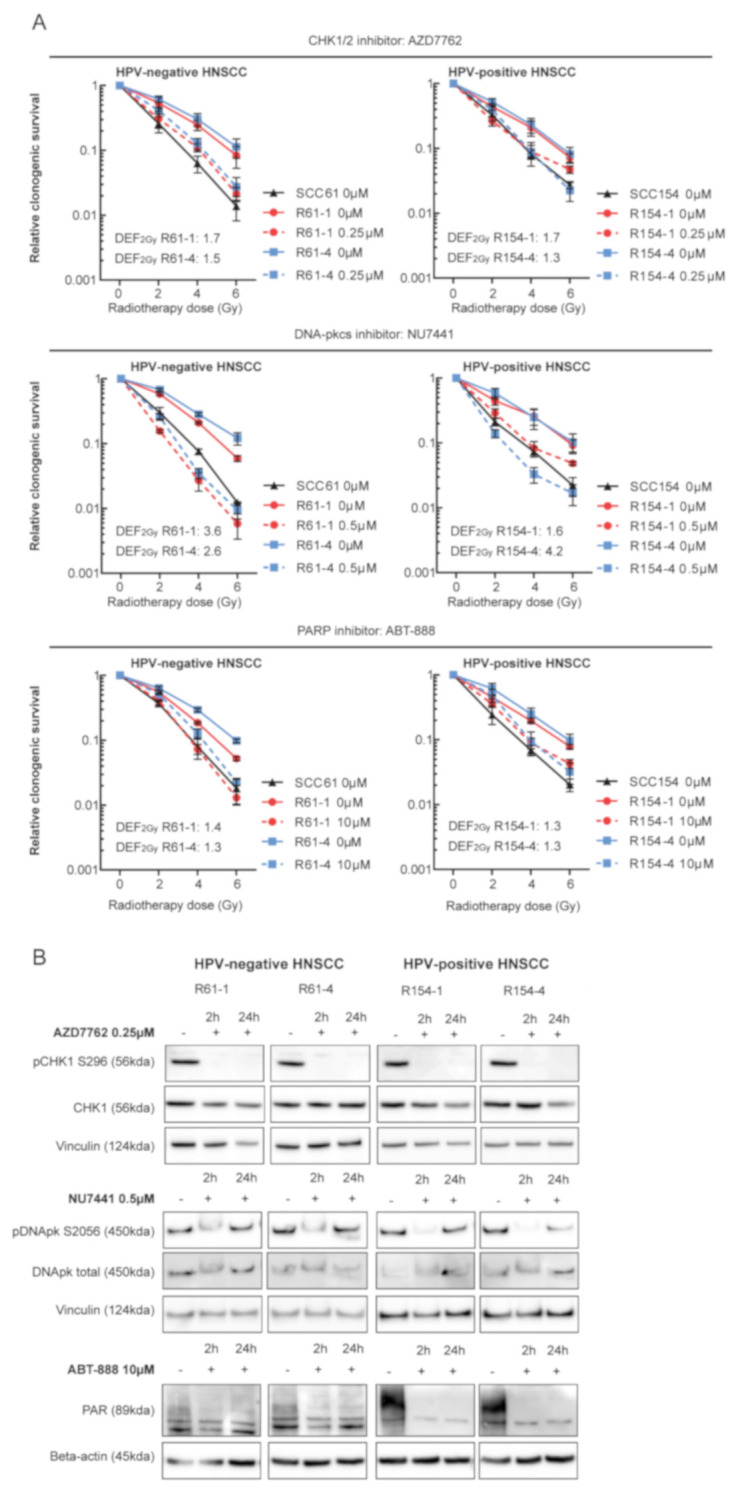
DDR inhibition in combination with RT radiosensitized HPV-positive and HPV-negative radioresistant HNSCC. (**A**) HPV-negative and HPV-positive radioresistant clones were pretreated for 2 h with DMSO (0 µM) or with indicated concentrations of DDR inhibitors and irradiated with the indicated doses. Medium was refreshed 24 h after drug exposure. Relative clonogenic survival was determined and is shown as the mean ± SEM clonogenic survival fraction of RT treated conditions relative to vehicle treated control cells, *n* = 3. (**B**) HPV-negative and HPV-positive radioresistant clones were treated with DMSO (−) or with indicated concentrations of DDR inhibitors. Cell lysates were prepared after 2 and 24 h of drug exposure. Immunoblotting of the following proteins was performed: pCHK1 S296, CHK1, pDNApk S2056, DNApk total, PAR, Vinculin and Beta-actin. KDA is the molecular weight as determined by the protein standard.

## Data Availability

The data presented in this study can be made available upon request from the corresponding author.

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
