# Peer review of "The DNA Damage Response Is Differentially Involved in HPV-Positive and HPV-Negative Radioresistant Head and Neck Squamous Cell Carcinoma"

_cancers, 2021, doi:10.3390/cancers13153717_

Round 1

Reviewer 1 Report

This work proposes the differences in the involvement of DNA damage response in radioresistance between HPV- and HPV+ cells.

The major problem is that the entire interpretation of the data is based on single-cell line from each group. THIS must be highlighted in the limitation of the study.

Major points:

Figure 1. Please show side-by-side the irradiated and non-irradiated groups (TVD graph). Also in the supplementary, show the graph of the actual tumor volume

Figure 2A, please show in the graph the data per cell and the distribution among all measured cells, not bar graph

Figure S2. it is not very clear to show only one side of the coin. The entire experiments shown in figure S2 must be assessed on the two models to make definitive conclusions.

Figure 2B the differences are not shown clearly with no statistic. Please consider other data representation

Consider strengthening the work by validating Figure 3A with multiple cell lines.

Author Response

Reviewer 1

  1. The major problem is that the entire interpretation of the data is based on single-cell line from each group. This must be highlighted in the limitation of the study.

Previous studies investigating the radioresistance phenomenon seen after fractionated irradiation with isogenic radioresistant cells have been focused on HPV-negative HNSCC. Since HNSCC are divided in two major groups (HPV-positive and HPV-negative) and these two groups also differ in molecular biology and response to RT, we decided to include cell lines from both HPV-positive and HPV-negative HNSCC. We were able to generate radioresistant cells from both groups and have included these in our results. However, we agree with the reviewer that multiple cell lines would increase the scientific relevance and robustness of our results and is indeed a limitation to our study. The limitation of using a single cell line from each group is highlighted in the discussion of the manuscript.

Major points:

  1. Figure 1. Please show side-by-side the irradiated and non-irradiated groups (TVD graph). Also in the supplementary, show the graph of the actual tumor volume

As requested by the reviewer side-by-side TVD graphs of irradiated and non-irradiated groups are shown in revised Figure 1B and C. In addition, graphs of the actual tumor volumes have been added in revised Figure S1B. These changes have been indicated in the result section of the revised manuscript.

  1. Figure 2A, please show in the graph the data per cell and the distribution among all measured cells, not bar graph

As requested by the reviewer, we have added the number of γH2AX foci per cell and the distribution among all measured cells (see revised Figure S2A and S3A for HPV-positive and HPV-negative HNSCC respectively in the revised manuscript). γH2AX foci imaging was performed with a widefield high-content imaging system (IN Cell Analyzer; GE life sciences) at 20x magnification. The analysis was performed in an automated manner with a standardized protocol using developer toolbox software (GE life sciences). For each condition and cell line, 3 independent experimental repeats were performed. Each experimental repeat consisted out of 2 technical repeats with a minimum of 1300 cell counts per technical repeat. Since the bar graph resulted in a clearer view of the γH2AX kinetics of the different cell lines, we opted to add the graphs with γH2AX foci per cell to the supplementary figures (revised Figure S2A and S3A).

  1. Figure S2. it is not very clear to show only one side of the coin. The entire experiments shown in figure S2 must be assessed on the two models to make definitive conclusions.

Taken into account the suggestion of the reviewer, we have performed additional experiments to present our results in an unambiguous manner for both HPV-negative and HPV-positive cells. More specific, we expanded the results of expression levels of important DDR proteins (previous Figure S2A) to HPV-negative HNSCC (revised Figure 2B.) These protein blots were moved to the main figure 2 (revised Figure 2B) in response to reviewer 4.

Figure S2B was a representation of G2/M phase of HPV-positive cells documented in Figure 2B (changed to Figure 2C in the revised manuscript). The representation was shown to highlight the faster G2/M arrest recovery after RT in HPV-positive radioresistant clones compared to parental cells. However, in response to point 5 of reviewer 1, we have changed the data representation of Figure 2C to investigate the differences in cycle distribution, including the G2/M phase, between HPV-positive and HPV-negative parental and radioresistant cells. Since the data is already presented in revised Figure 2C for both HPV-positive and HPV-negative HNSCC, we omitted the previous panel B of Figure S2.

Figure S2C, showed the CHK1/2 and CHK1 reliance’s of HPV-negative parental and radioresistant cells, we have expanded this data to the HPV-positive parental and radioresistant cells (revised Figure S6B and C). Although significant differences in different conditions (concentrations) were seen between HPV-positive parental cells and radioresistant clones, these differences were not consistent to form a conclusion.

Since both parental and radioresistant clones of HPV-positive cells showed an accumulation of G2/M arrest, showed no differences in S-phase regulation (revised Figure 2E; revised Figure S6B and C) and did not show differences in % of micronucleated cells, we did not opt to repeat the BRDU experiment in HPV-positive cells. The above-mentioned changes have been added to the results section of the revised manuscript.

  1. Figure 2B the differences are not shown clearly with no statistic. Please consider other data representation

To have a better statistical comparison between the parental and radioresistant cells, we have opted to change the data representation of Figure 2B (Figure 2C in the revised manuscript). More specific, we rearranged the cell cycle data to investigate the differences in cell cycle distribution at the indicated time points between parental and radioresistant cells via two-way ANOVA (Bonferroni multiple correction).

As previously stated, no statistical differences in cell cycle distribution can be seen between HPV-negative parental cells SCC61 and radioresistant clones R61-1 and R61-4. In contrast, radioresistant clones R154-1 and R154-4 clones show significantly less G2/M fraction at 32 until 72 hours after 6Gy when compared to the HPV-positive SCC154 cells. The changes in Figure 2C were added in the revised manuscript result section.

  1. Consider strengthening the work by validating Figure 3A with multiple cell lines.

The effect of key molecules of DDR inhibition have been investigated by several groups, including our group (references 11, 12, 15-17 in the revised manuscript) in a wide range of cell lines with different RT sensitivities. In this paper, we were interested in the effect of DDR inhibition on the RT response of radioresistant cells. Therefore, we opted to assess the clonogenic survival of the radioresistant clones upon targeted inhibition in combination with RT and compared their survival fraction to the survival fraction of their parental cells to investigate if the radioresistant phenotype could be rescued. We have clarified this also in the revised version of the manuscript.

Reviewer 2 Report

This is a nice and well structured analysis and manuscript which sheds light into the meaning of DNA damage response in radioresistence of HPV-positive and HPV-negative tumour cell lines. My personal impression of the study is very good and I would except for minors except the manuscript after only minor revision.

It appears to me that study design, methods, results and interpretation of results are sound and interesting. There only some minors I would like to address:

The minors are:

Could you, however, comment on how you verified radioresistance of the cell clones in more detail?

The simple summary is not a simple summary; no laymans speech

P1L7: double ;;

P1L55: bHPV?!

P2L47: locally advanced unresectable HNSCC....

In the discussion section and the abstract, I find the notion to jump from in vitro and in vivo xenograft models to some kind of treatment in humans a little farfetched. I would advise to tone that down a little. Additionally, I recommend to address the limitations of the study such as the tricky instance that only two cell lines have been investigated and on...

Author Response

Reviewer 2

The minors are:

  1. Could you, however, comment on how you verified radioresistance of the cell clones in more detail?

On first instance, the radiotherapy response of the generated clones was assessed on short-term by a Sulforhodamine B assay (SRB). For this, clones were seeded together with their parental cells in 96-well plates and after an attachment period of 24 hours, irradiated with doses of 0, 3 and 6Gy. One week after irradiation, all cells were fixed with trichloroacetic acid and stained with sulforhodamine B (Sigma-Aldrich, Saint-Louis, MO). Here after, 10mM Trisbase solution (Trizma, Sigma-Aldrich, Saint-Louis, MO) was added to release the SRB and absorption at 570nm was measured. Relative survival of each clone was determined compared to the unirradiated conditions (0Gy) using the optic density values. To define the increase in radioresistance, relative survival of the clones at 3 and 6Gy was compared to the relative survival of parental cells at 3 and 6Gy. Clones with an increase of relative survival of at least 1.2-fold and 1.8-fold at 3 and 6Gy respectively compared to the relative survival of parental cells, were considered as radioresistant clones.

After determining the radioresistant clones with SRB assays, the radiotherapy response of the selected clones was verified with the clonogenic assay, the gold standard for determining the radiotherapy response. The additional information is added to the material and method section of the revised manuscript.  

  1. The simple summary is not a simple summary; no laymans speech

As requested by the reviewer we adjusted to simple summary of the revised manuscript.

  1. P1L7: double ;;

P1L55: bHPV?!

P2L47: locally advanced unresectable HNSCC....

As requested by the reviewer we have adjusted these sentences in the revised version of the manuscript.

  1. In the discussion section and the abstract, I find the notion to jump from in vitro and in vivo xenograft models to some kind of treatment in humans a little farfetched. I would advise to tone that down a little.

As suggested by the reviewer, we attenuated the part in which the clinical translational potential is mentioned. See also revised manuscript: abstract and discussion for clinical translational potential of the in vitro models and clinical translational potential of the targeted DNA damage response inhibitors.

  1. Additionally, I recommend to address the limitations of the study such as the tricky instance that only two cell lines have been investigated and on...

As recommended by the reviewer, we added the limitation of our study in discussion section of the revised manuscript.

Reviewer 3 Report

In this manuscript, Bamps et al. look at the role of the DDR in the differences in survival after RT treatment observed between HPV+ and HPV- radioresistant head and neck squamous cell carcinoma cell lines. Whilst this is an important line of inquire, the data at this point seems rather preliminary.

Critically, the authors observe differences between the two radioresistant clones they created for each cell lin. They do not explain sufficiently why these differences occur and thus it would be pertinent to create additional radioresistant clones from additional HPV+ and HPV- HNSCC cell lines to provide more robust data.

  • Figure 2 - the authors must show representative IF images for yH2AX foci, representative flow cytometry plot for cell cycle data and  representative IF images for micronuclei in the main figure or as supplementary data.
  • How was survival determined in Figure 2D? The legend does not describe the method used.
  • Figure 2D - mitotic defects should also be investigated by looking at mitotic Histone H3 phosphorylation by WB or IF - micronuclei is not enough.
  •  The authors miss important references to important work in this area - e.g. Vitti et al., Cancers, 2020.

Author Response

Reviewer 3

  1. Critically, the authors observe differences between the two radioresistant clones they created for each cell lin. They do not explain sufficiently why these differences occur and thus it would be pertinent to create additional radioresistant clones from additional HPV+ and HPV- HNSCC cell lines to provide more robust data.

Previous studies investigating the radioresistance phenomenon seen after fractionated irradiation with isogenic radioresistant cells have been focused on HPV-negative HNSCC. Since HNSCC are divided in two major categories depending on HPV status and these two groups also differ in molecular biology and response to RT, we decided to include cell lines from both HPV-positive and HPV-negative HNSCC. We were able to generate radioresistant cells from both groups and have included these in our results. However, we agree with the reviewer that multiple cell lines would increase the scientific relevance and robustness of our results and is a limitation to our study. Since the generation of isogenic radioresistant clones is a labor-intensive process that takes more than a half year and from our recent experience we detected that not every cell line shows the potential to generate radioresistance, we opted to address this limitation in the discussion section of the revised manuscript.

  1. Figure 2 - the authors must show representative IF images for yH2AX foci, representative flow cytometry plot for cell cycle data and  representative IF images for micronuclei in the main figure or as supplementary data.

The representative immunofluorescence images for γH2AX foci (revised Figure S2B and S3B), flow cytometry (revised Figure S4) plots and representative images for micronuclei (revised Figure S5) were added to the supplementary data as suggested by the reviewer. In addition, references to these figures were added in the result section of the revised manuscript.

  1. How was survival determined in Figure 2D? The legend does not describe the method used.

Survival in Figure 2E (previously Figure 2D) was determined via clonogenic assay. Cells were seeded in limited densities and after attachment, treated with indicated concentrations of the inhibitors. Twenty-four hours after adding the drug, medium was refreshed and cells were able to form colonies. Colonies were fixed after 2-3 weeks and colonies containing 50 or more cells were counted. Relative clonogenic survival was determined for each concentration compared to the untreated (0µM) conditions for each cell line. We have added this information to the legend of the Figure 2E and adapted the Y-axis of the figure to relative clonogenic survival.

  1. Figure 2D - mitotic defects should also be investigated by looking at mitotic Histone H3 phosphorylation by WB or IF - micronuclei is not enough.

As requested by the reviewer, we reenforce our hypothesis about mitotic defects by assessing the histone H3 phosphorylation (pH3) (Supplementary Figure 6A). HPV-negative radioresistant clones R61-1 and R61-4 show a respective increase of 10.9- and 35.0-fold in pH3 expression levels compared to SCC61 cells 48h after RT. These results verify that radioresistant clones show an increase replicative ability after RT. Although lower compared to the RT treated conditions, the radioresistant clones show also higher pH3 expression at baseline when compared to the SCC61 cells. This could be explained by increased cell division in radioresistant conditions at baseline, which is in concordance with the doubling time data.

HPV-positive radioresistant R154-1 and R154-4 clones showed a 1.53- and 1.99-fold increase respectively in pH3 expression levels compared to the parental SCC154 cells, indicating an increase in mitotic cells. These results are in concordance with cell cycle distribution, with HPV-positive radioresistant clones showing faster cell cycle recovery. In unirradiated conditions, HPV-positive SCC154 cells showed higher pH3 levels compared to radioresistant R154-1 and R154-4 clones, suggesting that the parental cells have a higher fraction of mitotic cells in baseline conditions. However, we could not correlate these results with cell cycle distribution nor the doubling time, making it difficult to make firm conclusions. These addition have been added to the result section of the revised manuscript.  

  1. The authors miss important references to important work in this area - e.g. Vitti et al., Cancers, 2020.

As suggested by the reviewer we have added this Vitti et al. to our reference list (reference 17) in the revised manuscript.

Reviewer 4 Report

In the present study, the authors investigated the role of DDR in the radioresistance process of HNSCC by generating radioresistant clones from both HPV-positive SCC154 and HPV-negative SCC61 cells. The authors showed that HPV-positive and HPV-negative radioresistant clones are characterized by differential DDR response. Briefly, less residual DSBs damage and increased G2/M arrest recovery after RT was shown in HPV-positive radioresistant clones, whereas HPV-negative radioresistant clones showed less micronucleated cells after RT and increased survival upon checkpoint inhibition. I think that the topic and results of this manuscript is interesting. In my opinion, this manuscript after appropriate revision is suitable for the publication in Cancers.

Comments:

  1. Line 35: “bHPV”→ “bHPV”
  2. The information about X-ray machine used in this study is missing.
  3. Line 136: How did the authors selectively extract the nuclear proteins? The detail about it is missing.
  4. Line 164: The information about source of mice used in this study is missing.
  5. Supplementary figure 2A: Since this data is important data, please show it in the main text.
  6. To confirm the knockdown of target molecules, please add the western blot data of targets molecules in each knockdown cell as Supplementary figure.
  7. Fig. 3B: It seems that this data is n=1. Please increase sample size (at least n=3). 

Author Response

Reviewer 4

Comments:

1. Line 35: “bHPV”→ “bHPV”

As requested by the reviewer we have adjusted the sentence in the revised version of the manuscript.

2. The information about X-ray machine used in this study is missing.

As requested by the reviewer, we have added information about the used X-ray machines, Baltograph (199 kV photons, 15 mA, Balteau NDT) and Small Animal Radiation Research Platform (220 kV photons, 13 mA, Xstrahl), to the methods section of the revised manuscript. Both machines were used for performing the in vitro irradiation experiments. The small animal radiation research platform was used to perform the in vivo irradiation experiments.

 3. Line 136: How did the authors selectively extract the nuclear proteins? The detail about it is missing.

Proteins were extracted with RIPA buffer (150mM sodium chloride, 1% Triton X-100, 0.50 sodium deoxycholate, 0.10% SDS and 50mM tris pH 8.0). We did not perform separation of the nuclear fraction, we have clarified this in the material and method section of revised the manuscript.

 4. Line 164: The information about source of mice used in this study is missing.

As requested by the reviewer, we added the source of the mice to the material and methods section of the revised manuscript.

5. Supplementary figure 2A: Since this data is important data, please show it in the main text.

As requested by the reviewer, we moved Figure S2A to main Figure 2 (revised Figure 2B). In addition, we added these results to the results section of the revised manuscript.

6 and 7. To confirm the knockdown of target molecules, please add the western blot data of targets molecules in each knockdown cell as Supplementary figure.

Fig. 3B: It seems that this data is n=1. Please increase sample size (at least n=3). 

In a recent paper, we have performed a CRSIPR-screen targeting several DDR genes to assess the RT response in both HPV-positive and HPV-negative HNSCC. The screen showed us the importance of the genes involved in NHEJ, BER and cell cycle in the RT response. We validated the relevance of the effect of key DDR pathways in RT response with DDR inhibitors (see references 11 and 12 in the revised manuscript).

In line with this, we first assessed the CRISPR knockdown of the genes with a short-term survival assay and validated the effect of inhibition of key pathways using targeted DDR inhibitors with clonogenic assays. Since the limited time frame and the validation of the results with targeted inhibitors, we chose to move this data to the supplementary figures (revised Figure S7). We added the WB data of the DDR inhibitors as confirmation for the inhibition of the targets to Figure 3B of the revised manuscript.

Round 2

Reviewer 1 Report

In figure 1.

I expected to see side by side non-irradiated and irradiated per each sub cell line (for each one of them sepereratly Sensitive and two resistant)  , and therefore three small graphs per each cell line.   I could not  understand how the Chi was calculated and what was the comparison for.

Please correct, and good luck 

Author Response

In figure 1. I expected to see side by side non-irradiated and irradiated per each sub cell line (for each one of them separately Sensitive and two resistant)  , and therefore three small graphs per each cell line.   I could not  understand how the Chi was calculated and what was the comparison for. Please correct, and good luck 

All tumors were irradiated at an average tumor volume of 100mm³, the start of treatment was taken as day 1 for our tumor volume curves. This was the case for both the non-irradiated and irradiated conditions. However, the differences in baseline growth between different models (resistant clones and parental) resulted in differences in measurement days between different models. This made the statistical comparison between resistant and parental xenograft models difficult. To allow a side by side comparison of the RT response of resistant and parental xenograft models, we calculated tumor volume doubling (TVD) times for each condition. As mentioned in the manuscript, the TVD was defined as the time interval between the tumor volume at the start of treatment and the time when tumors reached twice their initial tumor volume. We used Kaplan-Meier method to test the differences in tumor volumes between the resistant and parental xenografts. Because the comparison was made between the 3 groups (2 radioresistant clones and parental cells) a chi-square test statistic was used. The test statistics indicates whether there is a difference between the three tested groups. We understand the request of the reviewer to see the RT response within each model and have provided the side by side K-M graphs of the irradiated and non-irradiated tumors for each model in the supplementary Fig 1B.

Reviewer 3 Report

The authors have significantly improved the manuscript based on my previous comments. I have only one remaining comment before publication.

The phospho-Histone H3 blot provided is not acceptable for publication and is not sufficient for to provide the quantification that the authors have provided. The authors should provide a better representative blot before publication so that the given quantification is believable.

Author Response

Review 3

The authors have significantly improved the manuscript based on my previous comments. I have only one remaining comment before publication. The phospho-Histone H3 blot provided is not acceptable for publication and is not sufficient for to provide the quantification that the authors have provided. The authors should provide a better representative blot before publication so that the given quantification is believable.

We agree with the reviewer that the quality of the phopsho-H3 blots were poor and quantification was difficult. We have reran the phospho-H3 blots for both HPV-positive and HPV-negative parental cells and their clones and replaced them in Figure S6A (see also revised manuscript).

Reviewer 4 Report

I am satisfied with the authors' responses.

Author Response

Review 4

I am satisfied with the authors' responses.

Thank you for your response.